# Transplantation of Predegenerated Peripheral Nerves after Complete Spinal Cord Transection in Rats: Effect of Neural Precursor Cells and Pharmacological Treatment with the Sulfoglycolipid Tol-51

**DOI:** 10.3390/cells13161324

**Published:** 2024-08-08

**Authors:** Alejandro Arriero-Cabañero, Elisa García-Vences, Stephanie Sánchez-Torres, Sergio Aristizabal-Hernandez, Concepción García-Rama, Enrique Pérez-Rizo, Alfonso Fernández-Mayoralas, Israel Grijalva, Vinnitsa Buzoianu-Anguiano, Ernesto Doncel-Pérez, Jörg Mey

**Affiliations:** 1Laboratorio de Regeneración Neural, Hospital Nacional de Parapléjicos, 45071 Toledo, Spain; aarrieroc@externas.sescam.jccm.es (A.A.-C.); searhe@hotmail.com (S.A.-H.); concepciongarciarama@gmail.com (C.G.-R.); jmey@sescam.jccm.es (J.M.); 2Facultad de Ciencias de la Salud, Centro de Investigación en Ciencias de la Salud (CICSA), Universidad Anáhuac México Norte, Huixquilucan 52786, Mexico; edna.garcia@anahuac.mx; 3Secretaría de la Defensa Nacional, Escuela Militar de Graduados en Sanidad, Ciudad de Méxcio 11200, Mexico; 4Instituto Mexicano del Seguro Social, Unidad de Investigación Médica en Enfermedades Neurológicas, Hospital de Especialidades, Centro Médico Nacional Siglo XXI. Av. Cuauhtémoc 330, Col. Doctores, Mexico City 06720, Mexico; stephanie.sanchez.torres@gmail.com (S.S.-T.); igrijalvao@yahoo.com (I.G.); 5Unidad de Ingeniería y Evaluación Motora del Hospital Nacional de Parapléjicos, 45071 Toledo, Spain; enriquep@sescam.jccm.es; 6Instituto de Química Orgánica General, CSIC, Juan de la Cierva 3, 28006 Madrid, Spain; alfonso.mayoralas@csic.es; 7EURON Graduate School of Neuroscience, 6229ER Maastricht, The Netherlands

**Keywords:** traumatic spinal cord injury, predegenerated peripheral nerve, neural precursor cells, Tol-51, axonal regeneration, modified BBB scale

## Abstract

Following spinal cord injury (SCI), the regenerative capacity of the central nervous system (CNS) is severely limited by the failure of axonal regeneration. The regeneration of CNS axons has been shown to occur by grafting predegenerated peripheral nerves (PPNs) and to be promoted by the transplantation of neural precursor cells (NPCs). The introduction of a combinatorial treatment of PPNs and NPCs after SCI has to address the additional problem of glial scar formation, which prevents regenerating axons from leaving the implant and making functional connections. Previously, we discovered that the synthetic sulfoglycolipid Tol-51 inhibits astrogliosis. The objective was to evaluate axonal regeneration and locomotor function improvement after SCI in rats treated with a combination of PPN, NPC, and Tol-51. One month after SCI, the scar tissue was removed and replaced with segments of PPN or PPN+Tol-51; PPN+NPC+Tol-51. The transplantation of a PPN segment favors regenerative axonal growth; in combination with Tol-51 and NPC, 30% of the labeled descending corticospinal axons were able to grow through the PPN and penetrate the caudal spinal cord. The animals treated with PPN showed significantly better motor function. Our data demonstrate that PPN implants plus NPC and Tol-51 allow successful axonal regeneration in the CNS.

## 1. Introduction

Traumatic spinal cord injury (SCI) frequently causes paraplegia with devastating personal consequences and significant economic costs to society. There is currently no cure for patients. The regenerative capacity of the CNS is limited by the formation of a physical barrier generated by the fibroglial scar [1] and contains growth inhibitory factors such as the myelin-associated proteins NogoA (neurite outgrowth inhibitor A), OMgp (oligodendrocyte myelin glycoprotein), and MAG (myelin-associated glycoprotein) [2,3]. These proteins bind to the Nogo receptor (NgR), which forms a complex with the p75 NGFR (p75 nerve growth factor receptor) on the cell surface and promotes signal transduction into the cytosol of neurons, activating the RhoA GTPase and ultimately inhibiting axonal growth [4,5]. In addition, extracellular matrix proteins such as sulfated proteoglycans with different glycosaminoglycans (GAGs) are secreted by reactive astrocytes and inhibit axonal growth, also by activating RhoA [4,6,7]. In response to injury, neurons activate signals such as growth-associated protein-43 (GAP-43) and microtubule-associated proteins to initiate axon regeneration [8,9,10,11]. However, the inhibitory microenvironment of the injured spinal cord prevents axons from regenerating after a lesion. Additionally, other signals, such as repulsive guidance molecule-A, semaphorin 3A, or ephrin B3, binding to their different receptors, inhibit axonal regeneration. Consequently, sustained axonal growth does not occur [12,13,14].

Few pharmacological therapies provide a solution in the chronic phase; the most promising approach is the reduction of fibroglial scarring [15]. Enzymatic inhibition of RhoA significantly enhances axonal regeneration and neural plasticity, leading to functional recovery in injury models. This provides a basis for the development of an effective treatment [3,7,16]. Tol-51 is a synthetic glycolipid that indirectly inhibits the glial scar effects of GAGs on the proliferation, growth, or migration of astroglia and microglia in cell culture because GTPases are involved in these processes. This is because Tol-51 inhibits the expression of the Arhgdia gene encoding RhoGDIα, which directly modulates the activity of RhoGTPases [17]. In a rat model of spinal cord injury, the application of Tol-51 in the acute phase was shown to favor axonal growth by inhibiting glial cells, thereby improving locomotor function [17].

Multiple treatments have been developed in recent years to minimize damage from injury or maximize regeneration. Most of these therapies focus on secondary events in the acute or subacute phase, but there is little research on injury repair in the chronic phase [18]. We hypothesize that at that stage, regeneration requires resection of the fibroglial scar [15,19]. This is the rationale for using combination therapies that reduce cellular damage, overcome inhibitory signals, and activate axonal growth to achieve long-distance axonal regeneration.

Cell therapies focus on their ability to reduce inflammation or provide regenerative signals for neurons. A class of neural progenitor cells, called aldynoglia (from the Greek, meaning “to make growth”), are a set of cells with immunophenotypic markers and common characteristics [20,21,22] that share characteristics with oligodendrocytes, astrocytes, and Schwann cells. These cells are prepared from the striatum of E15 rat embryos. Like Schwann cells, they have immunoreactivity for glial fibrillary acid protein (GFAP), S100β, and p75 NGFR [23]. Interacting with mature glia and neurons was shown to promote axonal growth and myelination [24]. In vitro studies showed that the aldynoglia cells have neural precursor cell (NPC) qualities such as high neural plasticity and the ability to migrate and premyelinate axons [25]. 

The key to promoting regeneration in chronic models is the use of bridges or scaffolds to promote a permissible substrate for axonal growth. Four decades ago, Aguayo and coworkers suggested transplanting peripheral nerves to allow axonal growth in the spinal cord [26]. Currently, predegenerated peripheral nerve (PPN) transplantation is considered an option for spinal cord repair. PPN acts as a neuroprotector in the spinal cord after transplantation. Axonal growth within the implant is supported by the permissive microenvironment derived from Schwann cells and macrophages present in the PPN. These present a pro-regenerative environment due to the presence of growth factors such as GDNF (glial cell-derived growth factor), NGF (nerve growth factor), NT-3 (neurotrophin-3), NT-4 (neurotrophin-4), and GM-CS (granulocyte and monocyte colony-stimulating factor) secreted by the PPN cells [27].

In addition, Schwann cells promote the myelination of regenerated axons [27,28,29]. After axonal growth through the PPN bridge has occurred, it is crucial to help the regenerating axons cross the interface into the spinal cord tissue at the rostral and caudal transplantation sites [30,31].

The aim of the present study was to improve procedures that support endogenous axonal regeneration and lead to functional recovery in a rat model of a complete spinal cord section. We suggest that the best strategy for this purpose consists of combining PPN implants with the following treatments. In order to reduce re-formation of the glial scar and improve axonal elongation, the RhoA inhibitor Tol-51 was used. For the continuous production of a growth promoting environment beyond the PPN implant, NPCs were injected locally.

## 2. Materials and Methods

### 2.1. Experimental Animals

All experiments followed European Council Directive No. 86/609/CEE and U.S. Department of Health guidelines to limit pain and discomfort to the experimental animals. The experimental protocol, surgical procedures, and postsurgical care were approved by the local Animal Welfare Ethics Committee of the Hospital National de Parapléjicos (HNP, number of registry 48-OH/2021) and ratified by the Department of Agriculture, Water and Rural Development of the Government of Castilla La Mancha (following the EU directive 2010/63/EU). All rats used in this study (E15 embryos and adult animals) were bred and maintained at the animal facility of the HNP.

### 2.2. Surgical Procedures

The study was performed with female Wistar rats (*Rattus norvegicus* with a body weight of 200–220 g) on the day of SCI. The standard housing conditions consisted of a 12 h light/dark cycle, 40–60% humidity, and 22 °C temperature with ad libitum access to food and water. A total of 62 rats were subjected to SCI. The animals were randomly divided into five groups: spinal cord injury followed by resection of the glial scar after one month (SCI), SCI with glial scar resection filled with alkaline fibrin (SCI+Alk-Fibrin), SCI plus replacement of the scar area with predegenerated peripheral nerve segments (PPN), PPN plus injection of Tol-51 (PPN+Tol-51), and PPN+Tol-51 plus injection of NPC (PPN+Tol-51+Cells). As no difference was observed between the SCI and SCI+Alk-Fibrin groups (*p* = 0.605), we decided to evaluate these groups statistically as a single control group (SCI) (Table 1). Twelve male rats were used as sciatic nerve donors, one for each SCI rat to receive PPN. Male rats were used as donors because the length of their sciatic nerve is longer than that of female rats.

### 2.3. Spinal Cord Transection

All surgical methodologies were developed in a previous study [30]. Sixty-two adult female rats (8–10 weeks old, 200–220 g) were used. Before SCI, all the rats were injected s.c. (subcutaneous) with 0.05 mg/kg buprenorphine (Dechra Ltd., Shrewsbury, UK). For spinal transection, the rats were anesthetized with 2.5% isoflurane (Baxter, Madrid, Spain) in oxygen at 0.5 L/min. Corneal dehydration was prevented with ophthalmic ointment (Lubrithal, Madrid, Spain). Body temperature was maintained using an electric heating pad. Following thoracic laminectomy at the T9 level, the dura mater was longitudinally opened to expose the spinal cord. Then, a complete transverse transection of the spinal cord was performed with microsurgical scissors. After stopping the bleeding, a microsurgical hook was used to ensure that the spinal section was complete, and the two separate spinal stumps were identified.

### 2.4. Predegenerative Peripheral Nerve for Transplantation

Twenty-one days before transplantation, 12 adult male Wistar rats were subjected to sciatic nerve injury (donor). Before the surgery, all the animals received buprenorphine (Dechra Ltd., UK) (0.05 mg/kg, s.c). The rats were anesthetized with isoflurane as for SCI, and the medial and posterior portions of the hind legs were sterilized superficially. Corneal dehydration was prevented with ophthalmic ointment, and the rats were placed on an electric heating pad. A longitudinal incision was made in the femoral side of each leg, and the sciatic nerves were severed with transverse sections near the proximal end of the femur. The distal stump of the nerve was sutured with 7-0 nylon to prevent spontaneous regeneration. On the day of transplantation, the rats were euthanized via i.p. injection of 100 mg/kg sodium pentobarbital, and the predegenerated nerves were extracted and placed in sterile PBS on ice until use.

### 2.5. Preparing Neural Precursor Cells for Transplantation

The isolation followed an established protocol [25]. E15 rat embryos were obtained by cesarean section from pregnant Wistar rats under deep anesthesia. Striata from E15 rats were dissected and mechanically dissociated into individual cells. The cells were seeded in a 75 cm^2^ culture flask with 15 mL of NB27 medium (neurobasal medium GIBCO, New York, NY, USA) mixed with B27 supplement (GIBCO 50×) supplemented with human 0.1% bFGF (of 10 ng/mL solution), 0.1% EGF (of 20 ng/mL solution, both from Peprotech, Cranbury, NJ, USA), 1% L-glutamate (of 200 mM solution, SIGMA, New York, NY, USA), 1% penicillin (of 5000 U/mL) and streptomycin (of 5000 mg/mL, GIBCO). Cultures were placed in a water-jacketed incubator at 37 °C with 5% CO_2_. After one week, when floating neurospheres had formed, they were collected by centrifugation, dissociated by mild trypsinization, and passed through a 25 G needle. The neurospheres were dissociated, and the cells were expanded every 3–4 days and used after the fourth passage for intraspinal transplantation.

### 2.6. Alkaline Fibrin and Tol-51 Sulfoglycolipid for SCI Treatment

The frozen components of Tisseel (Baxter, Spain), i.e., human fibrinogen (91 mg/mL) plus synthetic apothrombin (3000 IU/mL) and human thrombin (500 IU/mL) plus CaCl_2_ (40 μmol/mL), were gently thawed at room temperature, and separate aliquots were made and stored at −20 °C until use. To obtain alkaline fibrin, fibrinogen/apothrombin aliquots were gently thawed and diluted 1:1 with TBS (pH = 10), the same procedure used for spinal transplantation.

Tol-51 sulfoglycolipid was synthetized following the instructions previously described [32]. The identification of synthetic sulfoglycolipid at 550.3 (*m*/*z*) was monitored on a 4800 Plus MALDI TOF/TOF analyzer (Applied Biosystems, Toledo, Spain), as previously reported [33].

### 2.7. Transplantation Surgery

One month after complete transection, we performed the transplant surgery under the same conditions for anesthesia, eye hydration, and temperature, as mentioned above. The rats were randomly assigned to each experimental group. In all animals, the fibroglial scar was removed. The spinal cord was re-exposed in the area of the injury, and the scar tissue was identified and resected by cutting 1 mm rostral and caudal to the edges of the scar, leaving a cavity 7 mm in length. In the SCI group, this space was left empty. In the SCI+Alk-Fibrin group, the cavity following scar removal was filled with alkaline-modified fibrin. In the PPN group, the area was filled with six pieces of peripheral nerve (7 mm in length), which were bound with alkaline fibrin. In the PPN+Tol-51 group, two injections of 1 μL of Tol-51 (0.6 mg/kg each) were made into the rostral and caudal spinal cord stumps at lateral positions at a depth of 1 mm, and the cavity was filled as described for the PPN group. Finally, in the PPN+Tol-51+ NCP group, in addition to the treatments described above, two injections of 2.5 μL containing 10^6^ NPCs were administered. Both cellular injections were placed close to the CST at a depth of 1 mm, one into the rostral spinal cord segment and one into the caudal spinal cord segment.

### 2.8. Animal Care and Rehabilitation

To prevent dehydration, all animals were injected s.c. with 1 mL of isotonic saline. For one week, the rats received daily injections of 0.1 mL of marbofloxacin (5 mg/kg, s.c., antibiotic) (Farmacy, London, UK) and 0.15 mL of metacam (5 mg/mL, s.c., analgesic) (Proyma, Madrid, Spain). Food and water were provided ad libitum. Bladders and feces were emptied twice daily. In addition, to prevent intestinal problems and bladder infections, the rats were given a laxative (emuliquen laxante emulsion, 478.26 mg/mL + 0.3 mg/mL oral emulsion) for two months and one daily s.c. injection of buprenorphine (0.05 mg/kg) for one month. During the 3 months of evaluation, the animals underwent daily mobility exercises of the hind limbs to prevent spasticity, and twice a week, the animals were placed on a treadmill with elevation for 5 min and placed on a corrugated cardboard surface to stimulate the rehabilitation of the hips and hind limbs.

### 2.9. Open Field Test

Behavioral evaluation in the open field, once per week for twelve weeks, was carried out under red light to reduce stress. To assess improvements in hind limb locomotor function, a modified scale for transection animals, the modified BBB scale (mBBB), was used [31]. The rats were video recorded for five minutes/session, and the videos were analyzed by two blinded evaluators of the treatment conditions. Since most studies of SCI in rats, especially those involving milder lesions than complete transections, use the standard BBB scale [34], at 12 weeks after transplantation surgery, the BBB standard scale was assessed to prove differences between scales and compare results with other studies.

The scale consists of 22 items divided into 4 levels, which assess rhythmicity, alternation, and amplitude of movements and measure whether body weight support or plantar support is present [34,35]. Each level separates the degree of recovery that an animal with this injury can show. Level 1 is the absence of any movement of the three joints (knee, ankle, and hip); level 2 is the presence of movements of both hind limbs, which are rhythmic, in addition to the presence of plantar support with the dorsal part of the foot but without weight support; level 3 is the alternation of movements; and level 4 is the alteration of the movements, which assess rhythmicity, alternation, and amplitude of movements and determine the presence of body weight support or plantar foot placement [34,35]. 

### 2.10. Kinematic Analysis

For quantitative video analysis of limb movements, the skin of the animals was marked at reference points: the iliac crest, hip, knee, ankle, and fifth metatarsal of the hind leg. Rats were trained to move in a 100 cm × 30 cm walkway with transparent walls to be filmed with high-speed video cameras placed on both sides at 125 fps. At least three complete passes through the walkway were recorded for each rat once per month.

To analyze the kinematics, the videos were analyzed with the free software Tracker (Video Analysis and Modeling Tool. V 6.1.5), (https://physlets.org/tracker accessed on 1 July 2023) In this program, the recordings from both sides were synchronized, using a reference axis to calibrate the videos and obtain real distances. The normal (no SCI) and the injured rats gait were recorded. The linear (vertical and horizontal positions) and angular kinematic data (values of the angles of each joint) were entered into a software program, which was designed and programmed in Python by the Motor Engineering and Evaluation Unit of the HNP (Kinerat). The maximum and minimum angular values between the stance and swing phases were measured to obtain the angular amplitude, i.e., the amplitude of the flexion-extension of each joint, and the pendulum movement was obtained by measuring the swing of the movement between the hip and the fifth metatarsal. In addition, the distance between the hip and toe was measured, and the difference in the distance measured during the rats’ movements was used as another indicator of mobility.

### 2.11. Morphological Assessments

Twelve weeks after treatment, the animals were euthanized via i.p. injection of sodium pentobarbital (100 mg/kg) and intracardially perfused with 0.9% NaCl, followed by 4% paraformaldehyde. A 2 cm long segment of the spinal cord centered at the site of injury was removed and placed in 4% PFA for 24 h. Subsequently, the tissues were placed in PBS supplemented with 30% sucrose for 72 h. Next, the tissues were placed in OCT tissue freezing medium (Kaltek SRL, Padua, Italy), and 10 μm thick serial longitudinal sections were obtained with a Leica cryostat. Sections were evaluated with Masson’s trichrome light green staining following the instructions of the kit (Casa Álvarez, Madrid Spain). Myelin was visualized with the eriochrome cyanine (Sigma), followed by washing with 5% iron alum (Sigma) and a borax-ferricyanide differentiator (Sigma). Anatomical mapping and imaging were performed with an Olympus BX661 microscope with optical objectives: Plan N, 10×/0.25, and UPlanSapo, 20×/0.75.

### 2.12. Tracers to Assess Axonal Regrowth

Twelve weeks after treatment, axonal regeneration was evaluated with two different neurotracers, one for ascending fibers from the spinal cord and one for descending fibers from the motor cortex into the spinal cord. Anesthetized animals (atropine 0.05 mg/kg, pentobarbital 55 mg/kg, and xylazine 10 mg/kg) underwent stereotactic surgery by drilling the skull 2 mm anterior and 1.5 mm lateral to the bregma over both hemispheres. The tracer was injected through a microinjector (Kd Scientific 310-plus, Holliston, MA, USA) connected to a 5 μL Hamilton syringe. The tip of the syringe was placed on the brain surface and introduced 1.5 mm deep into the drilled canals. Four injections of 1 μL of dextran tetramethylrhodamine (Invitrogen, 3000 MW, Carlsbad, CA, USA) were given at a rate of 0.1 μL /min on each side, and the skin was sutured. Following this procedure, the spinal cord was exposed through a laminectomy performed at L3 for microinjection into the spinothalamic tract. Considering the central artery as a reference point, injections were made 1.5 mm lateral to each side and 0.5 mm deep. A total volume of 2 μL of the solution was administered at a rate of 0.1 μL/min of dextran fluorescein (Invitrogen, 3000 MW, Carlsbad, CA, USA). After 10 days, the animals were intracardially perfused with 4% PFA. The tissues were placed in OCT tissue freezing medium, and 10 μm thick serial parasagittal sections were obtained with a Leica cryostat. Sections were contrasted with Hoechst-33342 (dilution of 1:1000; Molecular Probes) and mounted with Immumount (Thermo Scientist, Madrid, Spain). The tissue was analyzed with an Olympus automated IX83 microscope (UPLXAPO 10×/0.40, objective) equipped with CellSens Dimensions software v.4.1.1 for automatic capture and mosaic composition of multiple images. Images were analyzed using QuPath Bioimage Analysis v.0.3.2 (https://qupath.github.io/). We counted all positive axonal profiles with neurotracers in bins of 250 μm along the complete section, from rostral to caudal for dextran tetramethylrhodamine, and from caudal to rostral for dextran fluorescein. Values were normalized to the axon counts before the entry point into the transplant.

### 2.13. Statistical Analysis

For the statistical analysis, GraphPad Prism V9.0.1 software (GraphPad, San Diego, CA, USA) was used, in which a D’Agostino and Pearson normality test was performed to determine whether the data had a normal distribution. All the results presented (BBB, mBBB, and kinematics) had a normal distribution. For the assessment of the BBB scale score at 12 weeks, a one-way ANOVA was performed, followed by Dunnett’s post hoc test. For mBBB, a two-way ANOVA was performed to compare the effects of treatment and time (weeks), followed by a post hoc Tukey´s test. For the kinematic data, a one-way ANOVA was used, followed by a Dunnett test for multiple comparisons, to determine differences with respect to the SCI group. For knee and hip plots, the data were processed using Python software. The time was normalized to the extension-flexion cycle, interpolating data where necessary. The significance level of the results was set at *p* < 0.05. The data are presented as the means ± SEMs. In the graphs, significance levels are indicated with asterisks: ** *p* < 0.01; * *p* < 0.05.

## 3. Results

### 3.1. Peripheral Nerve Transplantation after Complete Spinal Cord Transection

To assess axonal regeneration reliably, we made complete spinal cord transections at thoracic level T9. This is a very severe lesion, but it is considered necessary to test spinal reconnection with combinatorial therapy of biomaterials, cell transplantation, and pharmacological agents [30,36,37]. Despite intensive postoperative care, this disease model is characterized by high mortality, specifically during the first month after SCI (Table 1). No animals were lost due to the surgery per se, and after transplantation, there was no indication of sickness behavior or rejection of the transplanted tissue. In the following weeks, eight rats had to be sacrificed due to humane endpoints (urinary infection and self-mutilation). At the designated end of the study, 27 animals were evaluated (Table 1, Figure 1).

After transection, the rostral and caudal stumps of the spinal cord were separated, resulting in complete paralysis of the lower limbs (Figure 2A). One month later, a fibroglial scar had developed in the injury area (Figure 2B). This tissue was then surgically removed, leaving a cavity of ≈7.0 mm. In the respective treatment groups, intraspinal injections of sulfoglycolipid Tol-51 and NPC were made (Figure 2C), and the cavity was filled with the PPN graft and sealed with alkaline fibrin (Figure 2D).

### 3.2. Histological Changes in the Injured Area of the Spinal Cord and PPN Graft Zone

The transplanted PPN segments integrated into the recipient’s spinal cord without showing signs of host rejection in these treated groups, forming a robust bridge between the rostral and caudal injury zone (Figure 2E,F). At three months after transplantation, the grafts showed histological continuity with the host tissue at the rostral and caudal transition zones (Figure 2F). We observed in all the transplanted groups that the grafts had lost volume and were thinner than the adjacent spinal cord stumps (Figure 2E,F).

At the end of the study, Masson’s trichrome and eriochrome cyanine staining were performed to evaluate SCI tissue regeneration in the PPN grafts. In the group with SCI without an implant, the tissue showed extensive degeneration, and in the area of injury, a cyst separated both stumps of the spinal cord (Figure 3A). In the animals that underwent transplantation, the PPN tissue was preserved, and we found good integration of the graft. In addition, we observed fibers through the PPN bridge, which had a longitudinal alignment (Figure 3C). Eriochrome cyanine staining revealed preserved myelin in the rostral and caudal spinal cord segments of SCI rats without implants but not in the center of the injured rats (Figure 3B, blue). In the groups that underwent PPN transplantation, more myelin was preserved not only in the rostral and caudal spinal cord areas but also appeared within the PPN implants, indicating that regenerated fibers had been myelinated (Figure 3D). On the qualitative level, no differences due to treatment with NPC or Tol-51 were observed compared to those in the PPN group.

### 3.3. Analysis of Axonal Regeneration inside and beyond the PPN Grafts

A principal objective of this study was to study axonal regeneration into and beyond PPN implants. For this purpose, neurotracers were injected into the motor cortex and into the spinothalamic tract in the lumbar region. With cortical injection of tracers, axonal labeling (red) was detected in the rostral spinal cord of the implant zone, and with lumbar injection, axonal profiles (green) were detected in the spinal cord caudal to the implant (Figure 4). In rats with SCI but without PPN implantation, no labeled axons crossed the lesion site (Figure 4A,B), indicating that the growth of the tissue was not conducive to axonal regeneration. In all groups that underwent PPN transplantation, we observed labeled axons crossing the bridge and entering the caudal and rostral spinal cord areas with respect to the implant (Figure 4C–H).

To determine the additional effects of cellular and Tol-51 treatments, a quantification of axonal profiles with the tracers in three areas (rostral, implant site, and caudal zone) was performed, covering a region from 1 cm rostral to 1 cm caudal to the injury zone using 250 μm bins (Figure 5A,B). Compared with the SCI group, the PPN transplantation group had more axons labeled from the motor cortex in the rostral area. As mentioned above, labeled axons in the transplant zone were found only with PPN implants, and within the PPN, the number of labeled axons decreased with increasing distance from the implant/spinal cord apposition zone. The groups (PPN+Tol-51+NPC) and (PPN+Tol-51) had slightly more axonal profiles than the PPN-only group.

The efficiency of the tracer injections, indicated by the number of traced axons rostral to the epicenter area, differed among the rats. We therefore performed normalization using the numbers of labeled axons in the 5 mm segments caudal (for tracing from the lumbar spinal cord, green) or rostral (for tracing from the cortex, red) to the PPN implant (Figure 5C,D). With tracing from the motor cortex, the largest number of regenerating axons within the implant and in the caudal spinal cord area were found in rats treated with the combination of PPN+Tol-51+NPC. With tracing from the lumbar spinal cord, most regenerating axons appeared in the (PPN+Tol-51)-treated group. In all the rats, less than ¼ of the regenerated axons continued within the spinal cord tissue distal to the PPN implant. With only two rats per group for each combination of tracer and treatment, there were no significant differences due to pharmacological or cellular treatment, although the presence of a PPN implant was clearly necessary to induce axonal growth. In the PPN group, 10–30% of the axons penetrated the distal transition zones and reached 5 mm into the spinal cord tissue (Figure 5A–D).

### 3.4. Improvement of Functional Recovery in PPN Treatment Variants

Spinal cord transection caused complete paralysis of the hindlimbs. Three weeks after scar removal, the PPN group recovered leg mobility with occasional right/left alternations of movement. Over the next two months, the animals continued to improve. At 12 weeks, some rats achieved plantar placement of the feet and weight support while in a standing position. Since none of the rats reached the stage of weight-supported plantar stepping, the quantifiable changes in motor performance were confined to limb mobility. This was therefore quantified using kinematic video analysis (Figure 6). Angles at the articulations were continuously measured while the rats were moving through the passageway (Figure 6A–E). The black lines plotted in Figure 6D,E show the data of non-lesioned rats (mean with SEM) while walking along the passageway. In contrast, the rats with SCI, even after three months, dragged their hindlimbs and displayed much less mobility at the hip (Figure 6D) and knee (Figure 6E; green and gray lines). While PPN treatment did not significantly affect hip movements (Figure 6D), it resulted in larger amplitudes of leg movement measured at the knee joint (Figure 6E). Some transplanted rats were able to locate their feet under their bodies and gain weight support.

The principal evaluation of motor recovery in the open field was performed using the modified BBB scale (mBBB) [35]. For rats with complete spinal cord transection, we considered the mBBB scale more valid than the standard BBB scale because it takes the frequency of limb movements and the gradual improvement of hind limb mobility into account. This evaluation revealed that all the SCI groups with PPN implants exhibited continuous improvement in hind limb mobility during the three months of observation (Figure 7A, mBBB). This was not the case for the SCI-control group, in which recovery reached a plateau after two weeks [two-factor ANOVA; time: F (12, 433) = 66.5, *p* < 0.0001; treatment: F (3, 433) = 215, *p* < 0.0001; interaction: F (36, 433) = 5.08, *p* < 0.0001]. Post hoc Tukey tests demonstrated significant differences between all PPN-treated groups and the SCI group without implants (*p* < 0.001) from the third week onward. There were no effects of additional pharmacological or cellular treatment (*p* > 0.1 at all time points). To compare our results with published BBB data from other studies, we also analyzed the behavior of animals with a normal BBB scale (Figure 7B). Compared with the rats without implants, the rats in the PPN and combined treatment groups performed significantly better, while additional treatment with NPC or Tol-51 was not beneficial [one-way ANOVA, F (3, 23) = 6.26, *p* < 0.05; post hoc Tukey tests]. In addition to validating the kinematics analysis (see below), these data demonstrate the applicability of the mBBB scale for severe SCI models.

Once per month, a quantitative kinematic analysis was performed. As all kinematic parameters were severely reduced after spinal cord transection and the question of interest was the effect of treatment, the data from noninjured animals were not included in the ANOVA. At the end of the study, improvements were found to be greater in some PPN-implanted groups than in the SCI control group (Figure 7C–F). These differences were significant for mobility at the hip [Figure 7C, ANOVA, F (3, 18) = 4.3, *p* < 0.01] and knee [Figure 7D, ANOVA F (3, 18) = 5.4, *p* < 0.01] and not significant for leg movement in the anterior–posterior direction [pendulum, Figure 7E, ANOVA, F (3, 17) = 2.17, *p* = 0.13]. The movement amplitude between the hip and toe in the step cycle was also greater after PPN implantation [Figure 7F, ANOVA, F (3, 16) = 8.2, *p* < 0.01]. In the graphs, post hoc comparisons with the SCI group are indicated with * *p* < 0.05 and ** *p* < 0.01. We did not observe statistically significant differences between the different PPN groups. Thus, in accordance with the open field evaluation, the analysis indicated that PPN treatment was associated with more extensive hind limb movement and that neither Tol-51 nor the injection of NPCs produced an additional effect.

## 4. Discussion

Biomedical research has focused on identifying effective treatments for spinal cord injury, although to date, these treatments have not been successful. One of the problems is secondary degeneration, which causes cell death [38]. A no less important issue is the failure of surviving nerve cells to regenerate and make functional connections due to inhibitory factors, local inflammation, cyst formation, glial scars, and degenerating myelin [37]. At the clinical level, the greatest challenge consists of treating chronic patients where an inhibitory environment has already formed near the injury site [18,39,40]. For this reason, we chose the approach of removing scar tissue and replacing it with the pro-regenerative environment of a PPN and applying this treatment in the chronic stage after SCI. We also hypothesized that a combinatorial treatment to reduce re-formation of scar tissue and promote axonal growth would improve the results. This was carried out by the injection of the RhoA inhibitor Tol-51 and the implantation of NPCs that secrete growth factors. 

Our study shows that the implantation of PPN, even in the chronic phase after complete spinal cord transection, promotes axonal regeneration of the descending and ascending fiber tracts into and beyond the implant. Although further growth into the spinal cord tissue was limited, a significant functional improvement could be observed. However, we did not achieve additional benefits from the pharmacological and cellular treatments. The question arises as to whether other cell types would have been more efficient. A large number of different cellular therapies have been tested alone or in combination with pharmacology or biomaterials [39]. Schwann cells were already present in the implanted PPN. Stem cells derived from the bone marrow [30,40], umbilical cord, or adipose tissue [41] should be considered to improve the PPN implantation approach.

An additional issue is the transfer of the present results to the clinic, where complete spinal cord transection is, fortunately, a rare condition. The animal model with complete transection was chosen to unequivocally demonstrate axonal regeneration and differentiate it from the collateral sprouting of surviving fibers [42]. While this will not be translated directly to the clinic, as a proof of concept, the PPN implantation method can be applied to chronic SCI patients, where scar tissue may be surgically removed [15].

### 4.1. Axonal Regeneration

Despite the intrinsic ability of CNS axons to regenerate [5,43], the formation of the glial scar is considered an important factor in the inhibition of axonal growth [15]. Therefore, we selected a strategy based on implants of allogeneic predegenerated peripheral nerves as an oriented bridge. The first studies on SCI transplantation involving peripheral nerves were performed in the 1980s by Aguayo and coworkers, who demonstrated that fresh PN grafts serve as a favorable substrate for axonal growth in a spinal cord injury model [26]. Côté et al. (2011) demonstrated that predegenerated peripheral nerves favored neuroprotection of the tissue in chronic models of spinal cord injury and supported the growth of ascending tracts across the bridge to the caudal side of the spinal cord [27,44]. Previously, we demonstrated that predegenerated peripheral nerve tissue forms a good apposition with the spinal cord tissue after grafting. Rejection of the implants was not observed, and the PPNs presented a permeable microenvironment for axons [30,31]. The present results confirmed this finding, as we found that PPNs served as good substrates for long-distance regeneration of ascending and descending tracts. Fibers were able to penetrate the graft/spinal cord interface and grow into the anterior and posterior spinal cord stumps.

The regenerative properties of PPNs are attributed to their microarchitecture and to the presence of Schwann cells, macrophages, and connective tissue cells, which secrete trophic factors [45]. These trophic factors, including BDNF, GDNF, NT-3, NT-4, IGFs, and GM-CSF, support cell survival and axonal growth [27,46]. The basal lamina produced by Schwann cells serves as a proliferative zone where bands of Büngner are formed, which allows axons to grow in a linear and structured manner [45,46,47]. Although recent studies in SCI animal models [48,49,50,51], one clinical investigation [52], and much experience with peripheral nerve repair [53] indicate the promise of PPN transplantation, there are not enough studies of this strategy as a therapy for SCI.

To increase axonal growth in PPN grafts, we used combinations of the sulfoglycolipid Tol-51 and neural precursor cells. Tol-51 has been proposed to favor axonal growth by inhibiting the cells that form the glial scar [17]. The drug interacts with the protein RhoGDIα in the cytosol of the cell and acts as a regulator of RhoGTPases, which are involved in the RhoA signaling pathway and participate in the inhibition of axonal growth after injury. The sequestering of RhoGDIα by the sulfoglycolipid Tol-51 favors the ability of BDNF, which is secreted into the microenvironment, where it favors axonal growth and cell survival by activating the receptor TrkB in neurons [17].

The results showed no significant differences between the group with PPN grafts alone and the PPN+Tol-51 and PPN+Tol-51+NPC groups. This could be due to the severity of the SCI transection and treatment in the chronic phase. Treatment with Tol-51 might have been too late to produce cytoprotective effects. This explanation is plausible because it was previously reported that Tol-51, when injected into the spinal cord in the acute phase after contusion SCI, improved locomotor recovery in rats [17]. Based on the observation that Tol-51 reduces the activation of astrocytes [17], we investigated whether Tol-51 reduced glial scar formation after SCI. Since axons penetrated the transition zone between the implant and spinal cord tissue in all the PPN treatment groups, the formation of an inhibitory scar may not have been the limiting factor in this model. Inhibition of RhoA signaling is also expected to directly improve axonal regeneration [54,55]. In our study, regenerating axons penetrated the spinal cord tissue but grew no longer than 5 mm and thus failed to make contact with long-distance targets, such as lumbar motor neurons. It is possible that the injections of Tol-51 only at the time of scar removal were not sufficient to elicit an effect on axons several weeks later, when the axons had grown through the implant. Despite the negative evidence found in the present study, we therefore conclude that additional investigations of the role of NPC and Tol-51 in CNS repair are advised.

The second strategy to improve the therapeutic effect of PPNs is the use of NPCs. In complete section models, these cells have been demonstrated to favor axonal regeneration and the formation of synaptic connections [56,57,58,59]. These benefits were explained by Kitagawa et al. (2022), who proposed three hypotheses: (1) NPCs favor the reconstruction of neuronal circuits, either by supporting the reconnection of damaged axons with their original target or by the formation of new neurons that connect the damaged axons with neighboring axons or target neurons. (2) Another effect is the remyelination of axons due to the ability of NPCs to differentiate into oligodendrocyte precursors or Schwann cells. 3) Finally, the cells may have a neuroprotective effect due to the production of growth factors BDNF, GDNF, bFGF NT-3, NT-4, and VGF, which help to reduce secondary degeneration and support axonal growth, angiogenesis, and cell survival [60]. We can only speculate whether the NPC preparation used in this study (aldynoglia cells) was less effective than NPCs produced by others who observed axonal regeneration and functional connections [56,61]. In our previous publication (Doncel et al. 2009), we indicated that aldynoglia shares characteristics with glial cells such as GFAP, S100b, and p75 NGFR and thus could act similarly to astrocytes in the spinal cord [25]. More research is needed to understand the effect of Tol-51 on these NPCs, as this drug may have altered their biological effects in the context of SCI.

### 4.2. Functional Recovery

For the analysis of locomotor function, two tests were used: the modified BBB scale and kinematic analysis. All rats that underwent PPN transplantation recovered locomotor mobility of 4–9 points on the mBBB scale (level 2), which was not the case for the SCI control animals. From weeks 6–12, the mBBB of the PPN+Tol-51+NPC group improved by 8–10 points (level 3), indicating rhythmic and alternating movements with dorsal plantar support. This was more than what was observed in the other groups. Published studies with transection models in the acute phase reported that the use of treatments such as polyethylene glycol, cell transplants, or grafts of biomaterials or tissue improved locomotor function from the third week onward. At best, levels 8 to 10 of the standard BBB scale were obtained [62,63,64]. We confirmed our mBBB data with additional evaluations according to the BBB scale. The mean BBB permeability in the PPN+Tol-51 group reached 6.5 at 12 weeks after transplantation, which is slightly lower than that reported in the studies cited above. One of the weaknesses of open field evaluation (mBBB or BBB) is the lack of quantitative analysis of the amplitudes of limb movements. For this reason, we decided to perform a kinematic analysis of hip, knee, ankle, and pendular movements [65]. The results obtained with the PPN transplant groups showed wide movements of the three joints in both hind limbs in contrast to the control group. These data corroborate the results obtained in the open field. The fact that this treatment caused a low but significant functional improvement in comparison to SCI without the PPN suggests that some axons from regenerating neurons in the motor cortex made a functional connection.

## 5. Conclusions

Complete spinal cord transection causes paralysis, a high rate of mortality, and is not followed by functional recovery. The present results demonstrate that PPN implantation after scar removal one month after SCI induces the regeneration of descending and ascending axons within and beyond the PPN, which connects the rostral and caudal spinal cord segments. Behavioral evaluation revealed limited but significant motor recovery in the treated rats even after complete spinal cord transection. We consider this to indicate the functional reconnection of regenerated axons. We believe that combined treatment with PPN grafts is a suitable approach for investigating axonal regeneration in the subacute or chronic phase after SCI. Since the application of Tol-51 and NPC did not produce additional therapeutic benefits, other interventions are needed to promote functional regeneration in the chronic phase after SCI.

## Figures and Tables

**Figure 1 cells-13-01324-f001:**
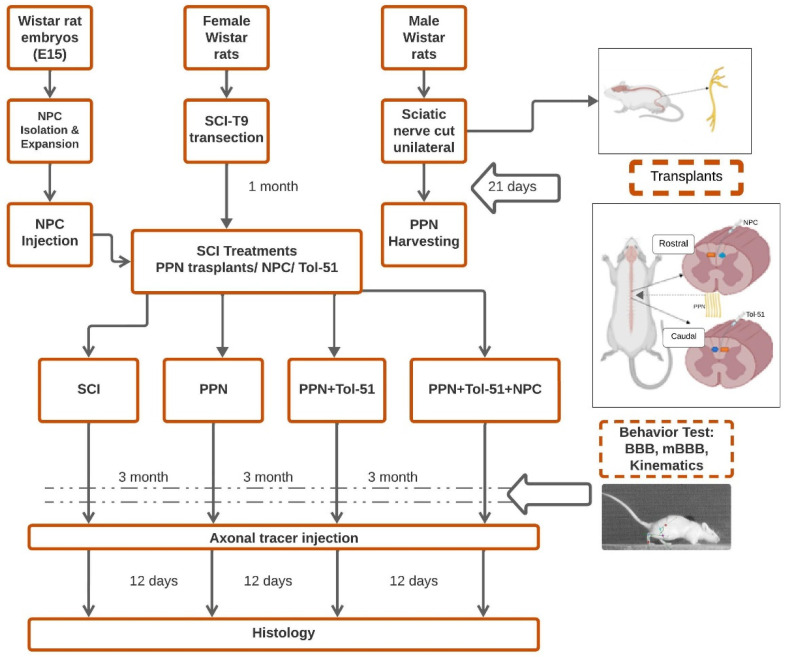
Experimental design for combination therapy after total transection of the spinal cord in rats. The study involved female Wistar rats undergoing surgical treatment one month after complete T9 spinal cord transection. Experimental therapy included the use of PPN implants along with pharmacological and cellular treatments. PPN implants were prepared using sciatic nerves from 12 male Wistar rats, which were transected 21 days prior to transplantation. NPCs were obtained from E15 rat embryo striata and expanded in vitro. One-month post-SCI, the SCI rats were randomly assigned to the following groups: the SCI (control), PPN, PPN+Tol-51, and PPN+Tol-51+NPC groups with different treatments. Sensory-motor recovery was assessed weekly in the open field, and kinematics at one and three months. After three months, axonal tracers were injected into the motor cortex and lumbar spinal cord, followed by histological processing 12 days later.

**Figure 2 cells-13-01324-f002:**
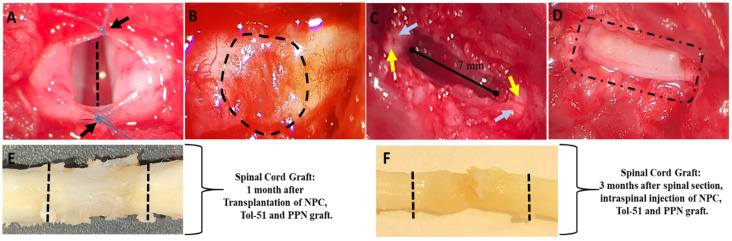
Images of complete spinal cord sections and images of nerve grafts. Following laminectomy and complete axial transection of the spinal cord at the T9 level ((**A**), discontinuous line), the dura mater was separated with sutures, as indicated by the black arrows. One month after SCI, a fibrogial scar and edema formed, as indicated by the dashed line (**B**). The tissue (≈7 mm) rostro-caudal extension, including the complete fibroglial scar, was surgically removed, and NPC (yellow arrows) and the sulfoglycolipid Tol-51 (blue arrows) were injected into the subdorsal spinal region (**C**). The spinal cavity was filled with six segments of the PPN and sealed with alkaline fibrin (**D**). Macroscopic views of the spinal cord one month (**E**) and three months (**F**) after grafting; discontinuous lines indicate the location of the PPN graft.

**Figure 3 cells-13-01324-f003:**
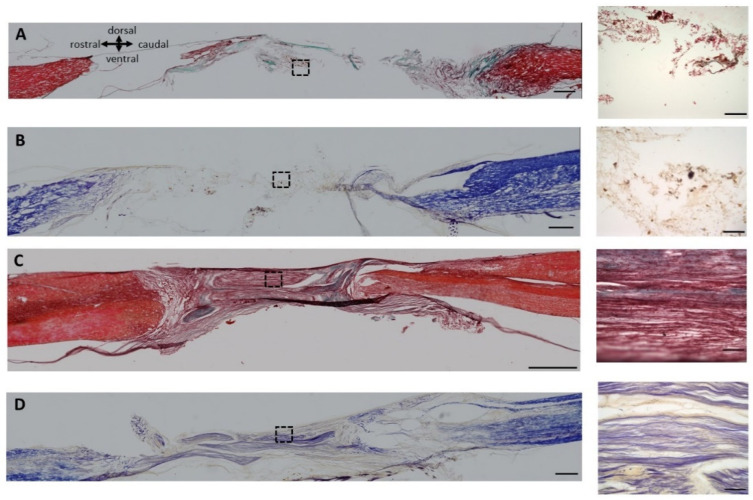
Histological analysis of the transected spinal cord with PPN grafts, which were fixed and stained three months after transplantation. Representative sagittal sections of 10 μm thickness from the transplanted spinal cord at the T9-T10 level. In the SCI control group (**A**,**B**) and in the (PPN+Tol-51+NPC) group (**C**,**D**), high-magnification images of the center of the lesion are shown, as indicated by the dotted squares in the right panel. Masson’s stains show a consolidated insertion of the PPN in the spinal cord 3 months after the graft, with high production of fibers (**C**) and alignment of these fibers (see the respective magnification area). In control SCI sections, only few disorganized fibers were found (**A**). Eriochrome staining (**B**,**D**) revealed blue staining for myelinated fibers in the PPN+Tol-51+ NPC transplants (**D**), which was not found in the spinal tissue of the untreated SCI group (**B**). The scale bars in the high-magnification images are 500 μm (**A**–**D**) and 100 μm.

**Figure 4 cells-13-01324-f004:**
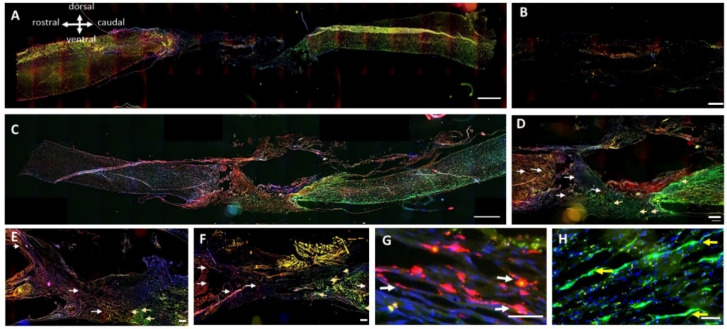
Axonal tracing of regenerating axons three months after grafting. The red tracer (dextran-tetramethylrhodamine) was injected into the motor cortex, and the green tracer (dextran-fluorescein) was injected into the spinothalamic tract (L3 level) behind the caudal end of the transplants; cell nuclei were counterstained with Hoechst-33342 (blue). Representative composite images of T8-L2 spinal tissue containing the areas of injury in rats with SCI, the control group (**A**,**B**) and the SCI+PPN group (**C**,**D**). The graft areas for the groups treated with (PPN+Tol-51) and (PPN+Tol-51+ NPC) are shown in E and F, respectively. Higher magnification images of the graft areas reveal new axonal projections within the implants (**G**,**H**). Axons traced from the motor cortex grew within the graft toward the caudal segments of the spinal cord (white arrows in (**D**–**G**)). Axons that were traced from the grafted lumbar spinal cord grew toward the rostral segments of the spinal cord (yellow arrows in (**D**–**F**,**H**)). Scale bars at 1000 μm in A, B; 500 μm in (**C**–**F**); and 250 μm in (**G**,**H**).

**Figure 5 cells-13-01324-f005:**
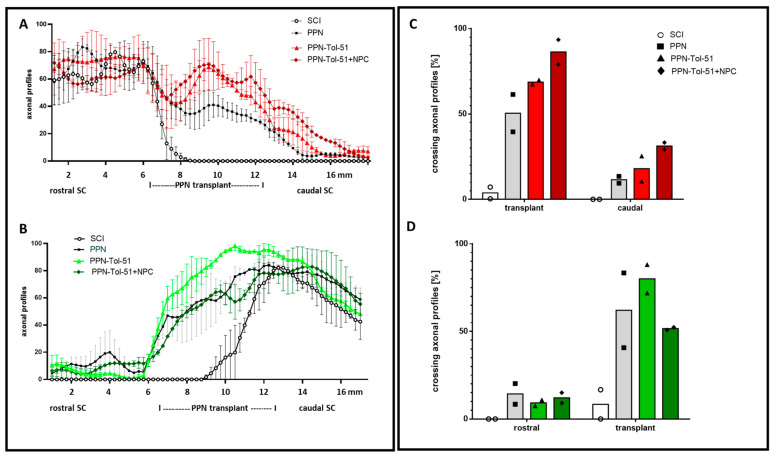
Quantification of axonal regeneration in the SCI group and PPN graft groups. In the left panel, the graphs show the percentage of tracer-labeled axons in the sections counted at 250 μm intervals; mean +/− SD and *n* = 2 rats/group (**A**,**B**). Axons were labeled with tracer injection from the motor cortex ((**A**), red) and from the lumbar spinal cord ((**B**), green). Regenerative axonal profiles were found within the PPN implants but not in the corresponding area of the SCI group. In panel (**C**), the number of regenerating axons within the transplant site and in the spinal cord caudal to the injury site was normalized to the number of labeled axons rostral to the injury site (i.e., in the rostral segment of the spinal cord for tracing axons from the cortex area). In bar chart (**D**): Quantification of axonal regeneration with axonal tracing of the lumbar spinal cord (using trace data from the caudal section of the spinal cord for normalization). A minority of regenerating axons found within the PPN implants continued to grow into the distal segments of the spinal cord. This percentage was greater for treatment with (PPN+Tol-51+NPC) for the anterior tracer (**C**) and for treatment with (PPN+Tol-51) for the posterior tracer (**D**). The columns show the average of two rats/group.

**Figure 6 cells-13-01324-f006:**
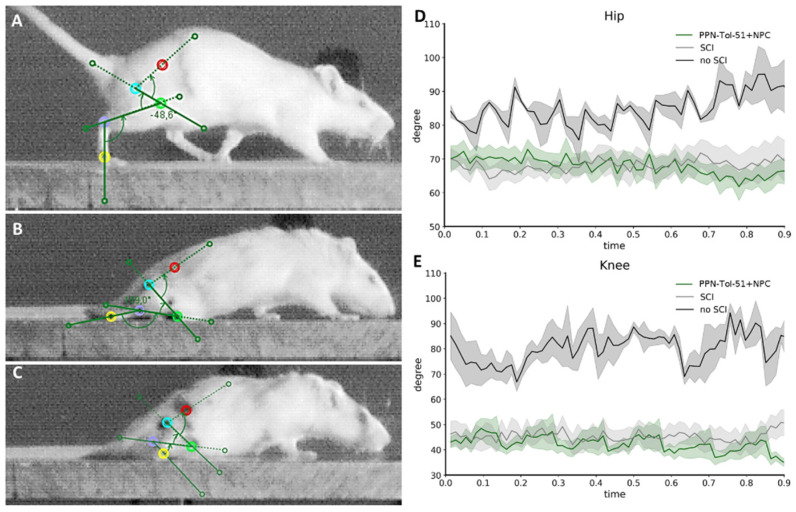
Kinematic analysis of motor functions in the SCI group and the PPN+Tol-51+Cells group. Video frames showing angle measurements at the hip, knee, and ankle (green lines) during a rat’s movement along a passageway (**A**–**C**). For the analysis, the iliac crest (red), hip (blue), knee (green), ankle (purple) and toe (yellow) of the animals were marked. Rats without lesion (**A**), with SCI and scar removal but without PPN implantation (**B**) and treated with PPN+Tol-51+NPC (**C**) are shown. Original kinematic data are shown in panels (**D**,**E**). The plots show changes in the hip (**D**) and knee (**E**) angles, representing the average of all rats in the SCI group, the PPN+Tol-51+NPC group and those without SCI. Shaded color represent the SEM.

**Figure 7 cells-13-01324-f007:**
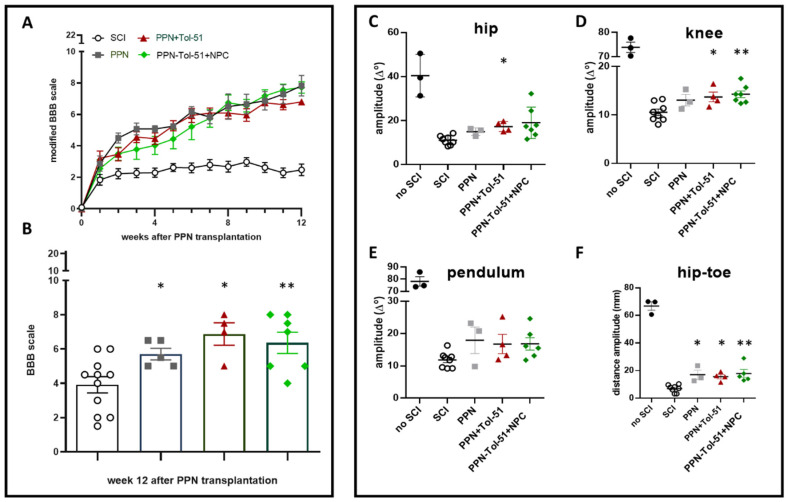
Recovery of motor functions after SCI and PPN graft variants. Evaluation in the open field: Modified BBB scores were monitored once per week starting with treatment at one month after SCI (**A**). The effects of time after treatment and mode of treatment were highly significant, with the PPN transplants being therapeutically effective without differences due to treatment with Tol-51, NPC, or both; evaluation in the open field using the mBBB (**A**) and BBB grading scale (**B**), quantitative evaluation of video data (**C**–**F**). While there were no significant differences between the groups that received PPN transplants, a significant treatment effect compared to that of SCI was observed only for PPN+Tol-51 with respect to the hip or knee amplitude angle (**C**,**D**), for PPN+Tol-51+NPC with respect to the knee amplitude angle (**D**) and for all PPN treatment groups with respect to the hip-toe distance amplitude (**F**). Refer to the main text for statistical evaluation. Significance levels compared to the SCI group are indicated with * *p* < 0.05 and ** *p* < 0.01. In all SCI groups, motor performance was strongly reduced compared with that of rats without spinal cord transection (no SCI).

**Table 1 cells-13-01324-t001:** Loss of experimental animals due to postsurgical complications and humane endpoints.

Experimental Groups	SCI	SCI+Alk-Fibrin	PPN	PPN+Tol-51	PPN+Tol-51+NPC
*n* (total = 62)	12	12	12	12	14
Animals lost after SCI:					
Before transplant	6	6	5	7	3
Animals lost after transplant surgery:					
Month 1	1	0	2	1	2
Month 2	0	0	0	0	2
Month 3	0	0	0	0	0
Month 4	0	0	0	0	0
Final group size:					
	SCI	SCI+Alk+Fibrin	PPN	PPN+Tol-51	PPN+Tol-51+NPC
*n* (total = 27)	5	6	5	4	7

## Data Availability

The datasets used and/or analyzed during the current study are available from the corresponding authors upon reasonable request.

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
