# Peer review of "Transplantation of Predegenerated Peripheral Nerves after Complete Spinal Cord Transection in Rats: Effect of Neural Precursor Cells and Pharmacological Treatment with the Sulfoglycolipid Tol-51"

_cells, 2024, doi:10.3390/cells13161324_

Round 1

Reviewer 1 Report

Comments and Suggestions for Authors

The authors assessed  axonal regeneration and locomotor function improvement after SCI in rats treated with a combination of PPNs, NPCs and Tol-51. The therapeutic effect of PPNs was not enhanced by NPCs or Tol-51. My only concerns are only related to the number of replicates: 

Figure 5 data are based on N=2 replicates. The authors should have a larger cohort to support their data. Also, it would be better to show the axonal profile across the length of the SC in panels A and B, using the values normalized for the number of traced axons.

Figure 6 shows only data from N=1. Why do not showing the average and variance of all the replicates? I would suggest adding more replicates to the analysis. 

Author Response

We thank the reviewer for their effort and comments which were very helpful in improving the manuscript. As detailed below, we have tried to implement all suggestions. All changes to the text are highlighted in yellow. A new Figure 6 containing data of all rats is being submitted, and Figure 5 replotted according to the reviewer’s suggestion.

Reviewer1

  1. The authors assessed axonal regeneration and locomotor function improvement after SCI in rats treated with a combination of PPNs, NPCs and Tol-51. The therapeutic effect of PPNs was not enhanced by NPCs or Tol-51. My only concerns are only related to the number of replicates:  Figure 5 data are based on N=2 replicates. The authors should have a larger cohort to support their data. Also, it would be better to show the axonal profile across the length of the SC in panels A and B, using the values normalized for the number of traced axons.
  2. We have revised and replotted Fig. 5 according to the reviewer’s suggestion. The legend has been adapted accordingly. Data are now normalized for the highest number of traced axons before the transplant using a gliding average. In complete agreement with reviewer we would have preferred a larger number of animals in the final histological evaluation. When planning the experiment the attrition rate, due to the complete spinal cord transection was underestimated. Nonetheless, the better axonal growth in the treatment groups compared to the controls is beyond doubt. Given the absence of statistical testing, we make no quantitative statement. Changes in methods and Figure 5 legend are emphasized in yellow.
  3. Figure 6 shows only data from N=1. Why do not showing the average and variance of all the replicates? I would suggest adding more replicates to the analysis. 
  4. Figure 6 has been revised according to the reviewer’s suggesting, and all data have now been incorporated in the graphs. The plots show averages of all rats/group with SEM a shaded area.

The reason that we showed only N=1 in Figure 6 is that this graph was only to show an example of the data whose quantification is presented in Fig. 7C-F. All changes in methods and Figure 6 legend are highlighted in yellow.

Reviewer 2 Report

Comments and Suggestions for Authors

In this work, the authors tested whether transplantation of predegenerated peripheral nerves (PPNs) after complete spinal cord transection in rats, at chronic stage of SCI, can result in partial axonal regeneration and motor function improvement. To use PPNs for axon regeneration after SCI, including chronic SCI, is a long-known concept dating back to 20th century. There are many works similar to this one, the only difference being combinational treatment: PPNs + neural progenitor cells, NPCs (to secrete neurothrophic factors) + sulfoglycolipid Tol-51 (to inhibit astroglyosis) used by the authors in this work alongside with PPNs only treatment and PPN + Tol-51 treatment. Thus, in my opinion the work lacks novelty. Moreover, as the authors state themselves, perhaps timing of Tol51 application was not the best one. There were no additonal therapeutic benefits of adding Tol-51 and NPCs to the PPNs. Please find my comments below.

Line 20. “the regeneration of CNS axons has been shown to occur by grafting predegenerated peripheral nerves (PPNs) and to be promoted by the transplantation of neural precursor cells (NPCs). The introduction of a combinatorial treatment of PPNs and stem cells after SCI ….

- The authors started from talking about NPCs and suddenly switched to talking about stem cells. Please be consistent.

What type of NPCs? Directly reprogrammed NPCs (drNPCs) perhaps? Aldynoglia? Aldynoglia derived from drNPCs? Neurospheres

(aldynoglia?) formed by embryonic cells? In this study the authors used the latter, but that about clinical applicability of their findings, what types of NPCs eventually should be used in clinic when treating patients?

When talking about Nogo receptor (NgR)-mediated RhoA GTPase signaling in context of neuroregeneration and axonal growth, it worth mentioning approaches aimed to modulate the aforementioned signaling pathways. The authors discuss it briefly in line 70, but I think this info belongs to the very beginning of introduction (~lines 43-50)

Line 51. “in the inhibitory microenvironment” - please fully describe the composition of such microenvironment, apart from GAGs previously mentioned in the text.

Line 86. “M2 macrophages” please explain briefly why M2 phenotype is important for neuroregeneration

Line 86. Speaking about “the presence of growth factors such as GDNF (glial cell-derived growth factor), BDNF (brain-derived growth factor), NGF (nerve growth factor), NT-3 (neurotrophin-3), NT-4 (neurotrophin-4), and GM-CS (granulocyte and monocyte colony-stimulating factor)” - what is expected to be a source of these factors? Cell therapy (cells secreting these factors)? What type of cells? Schwann cells??? Or NPCs? Perhaps delivery to the lesion site of recombinant growth factors (rGF) can be used as well. What carriers to be used in such a case to deliver rGF? Should a constant release from “carrier” or a single application be used, simultaneous addition or subsequent treatments? What concentration to use? Etc).

Line 96. “We suggest that the best strategy for this purpose consists of combining PPN

implants with pharmacological and cellular treatments” - please provide an example of pharmacological and cellular treatments (“We suggest that the best strategy for this purpose consists of combining PPN implants with pharmacological and cellular treatments, such as …..”)

Line 254 “connected to a 5 l Hamilton” - please fix the typo

Line 256. “Four injections of 1 L of dextran tetramethylrhodamine (Invitrogen, 3000 MW) were given at a rate of 0.1 l/min on each side” please fix the typo

Figure 1. Regarding an Experimental design flowchart – could you please make font of the text bigger?

The burning question is – are these data translatable to human subjects?

As for “ the failure of surviving nerve cells <in humans> to regenerate and make functional connections. This is due to inhibitory factors in the glial scar and degenerating myelin” , the one may argue that in so-called regenerating species the there are inhibitory factors and degenerating myelin too, but it's not a limitation in their case, so regeneration failure in humans may not be explained solely by these factors. Please, elaborate on this subject.

Line 503. “Tol-51 that favors axonal growth by inhibiting pathways such as RhoA” - why do not use a small molecule inhibitors or siRNAs instead?

Line 504. “promotion of the secretion of growth factors and cell survival through the use of neural precursors” - what about other types of cell therapy, apart from NPCs?

Given the fact that, as the authors state, the used in this study “type of lesion is not common in patients”, how applicable to clinic are the findings of the current work?

Having said all of this, all experiments were performed “according to the book”, a big study group of animals (sixty-two) was used in this work, which makes it statistically robust. At least 12 animals were allocated to each group (SCI, SCI + PPN, etc).

Perhaps expanding discussion about previous works on transplantation of PPNs in SCI would have strengthened the manuscript.

Author Response

We thank the reviewer for their effort and comments which were very helpful in improving the manuscript. As detailed below, we have tried to implement all suggestions. All changes to the text are highlighted in yellow.

Reviewer2:

  1. Line 20. “the regeneration of CNS axons has been shown to occur by grafting predegenerated peripheral nerves (PPNs) and to be promoted by the transplantation of neural precursor cells (NPCs). The introduction of a combinatorial treatment of PPNs and stem cells after SCI ….”- The authors started from talking about NPCs and suddenly switched to talking about stem cells. Please be consistent.
  2. We have eliminated the confusion and now use the term NPCs throughout the text.
  3. What type of NPCs? Directly reprogrammed NPCs (drNPCs) perhaps? Aldynoglia? Aldynoglia derived from drNPCs? Neurospheres(aldynoglia?)formed by embryonic cells? In this study the authors used the latter, but that about clinical applicability of their findings, what types of NPCs eventually should be used in clinic when treating patients?
  4. Following the reviewer’s comment, we now explain the NPC as aldynoglia cells that were prepared from the striatum of E15 rat embryos. We have revised the discussion considering the reviewer’s comments.
  5. When talking about Nogo receptor (NgR)-mediated RhoA GTPase signaling in context of neuroregeneration and axonal growth, it worth mentioning approaches aimed to modulate the aforementioned signaling pathways. The authors discuss it briefly in line 70, but I think this info belongs to the very beginning of introduction (~lines 43-50)

 R. We appreciate the suggestion and have amended the introduction accordingly.  Line 51. “in the inhibitory microenvironment” - please fully describe the composition of such microenvironment, apart from GAGs previously mentioned in the text.

  1. Again, additional information has been added: “Additionally, other signals, such as repulsive guidance molecule-A, semaphorin 3A or ephrin B3 binding to their different receptors, inhibit axonal regeneration. Consequently, sustained axonal growth does not occur [12–14]”.

 Line 86. “M2 macrophages” please explain briefly why M2 phenotype is important for neuroregeneration

  1. According to the reviewer comment, traditionally macrophage phenotypes are distinguished according to their phenotype, be it pro-inflammatory (M1) or rather pro-regenerative (M2). Since macrophages in the PPN promote regeneration we described them as M2 in the introduction. However, the concept describes a continuity of macrophage activation rather than just two phenotypes. Since we do not have data regarding these phenotypes in the PPN. Therefore, in the revision we do not mention M2.
  2. Line 86. Speaking about “the presence of growth factors such as GDNF (glial cell-derived growth factor), BDNF (brain-derived growth factor), NGF (nerve growth factor), NT-3 (neurotrophin-3), NT-4 (neurotrophin-4), and GM-CS (granulocyte and monocyte colony-stimulating factor)” - what is expected to be a source of these factors? Cell therapy (cells secreting these factors)? What type of cells? Schwann cells??? Or NPCs? Perhaps delivery to the lesion site of recombinant growth factors (rGF) can be used as well. What carriers to be used in such a case to deliver rGF? Should a constant release from “carrier” or a single application be used, simultaneous addition or subsequent treatments? What concentration to use? Etc).
  3. According to the reviewer comment indeed, re-differentiating Schwann cells in crushed PN have been shown to produce a range of neurotrophic factors. The use of PPN is justified precisely by the growth-promoting properties of the cells contained therein, without necessarily characterizing the molecular composition of these factors.
  4. Line 96. “We suggest that the best strategy for this purpose consists of combining PPN implants with pharmacological and cellular treatments” - please provide an example of pharmacological and cellular treatments (“We suggest that the best strategy for this purpose consists of combining PPN implants with pharmacological and cellular treatments, such as …..”)
  5. In response to the reviewer’s comment, the objective of the study has been rephrased to include the concrete information: “The aim of the present study was to improve procedures that support endogenous axonal regeneration and lead to functional recovery in a rat model of a complete spinal cord section. We suggest that the best strategy for this purpose consists of combining PPN implants with the following treatments: In order to reduce re-formation of the glial scar and to improve axonal elongation, the RhoA inhibitor Tol-51 was used. For the continuous production of a growth promoting environment beyond the PPN implant, NPC were injected locally.”

.

  1. Line 254 “connected to a 5 l Hamilton” - please fix the typo
  2. The error has been corrected.
  3. Line 256. “Four injections of 1 L of dextran tetramethylrhodamine (Invitrogen, 3000 MW) were given at a rate of 0.1 l/min on each side” please fix the typo
  4. The error has been corrected.
  5. Figure 1. Regarding an Experimental design flowchart – could you please make font of the text bigger?
  6. The suggested changes have been applied.
  7. The burning question is – are these data translatable to human subjects?As for “ the failure of surviving nerve cells <in humans> to regenerate and make functional connections. This is due to inhibitory factors in the glial scar and degenerating myelin” , the one may argue that in so-called regenerating species the there are inhibitory factors and degenerating myelin too, but it's not a limitation in their case, so regeneration failure in humans may not be explained solely by these factors. Please, elaborate on this subject.
  8. We agree with the reviewer’s point regarding the translation to the clinic. An additional consideration related to the clinical relevance has been added to the discussion. Indeed, spinal cord regeneration in amphibians and fishes is observed in the presence of degenerating myelin, though with differences in scar formation. We think that discussing these differences exceeds the scope of the present study.
  9. Line 503. “Tol-51 that favors axonal growth by inhibiting pathways such as RhoA” - why do not use a small molecule inhibitors or siRNAs instead?
  10. We agree now, after seeing our results, that the reviewer’s suggestion to use siRNA or other inhibitors of the pathway may be more efficient. This will certainly be considered in future experiments. We used Tol-51 because the molecule was effective in vitro and as an acute treatment after spinal cord contusion injury (García-Álvarez et al. 2015), and for this reason the molecule was patented by the group. Although the results were not as expected, we believe that Tol-51 for acute or sub-acute models could be a possible therapy for a pre-clinical study.
  11. Line 504. “promotion of the secretion of growth factors and cell survival through the use of neural precursors” - what about other types of cell therapy, apart from NPCs?
  12. The discussion has been changed according to the suggestion: “The question arises whether other cell types would have been more efficient. A large number of different cellular therapies have been tested alone or in combination with pharmacology or biomaterials (Tetzlaff et al. 2011). Schwann cell were already present in the implanted PPN. Stem cells derived from the bone marrow (Aguado-Garrido et al. 2024, Buzoianu et al. 2020) umbilical cord or adipose tissue (Liu et al. 2020) should be considered to improve the PPN implantation approach.”

Given the fact that, as the authors state, the used in this study “type of lesion is not common in patients”, how applicable to clinic are the findings of the current work?

  1. Even though in the clinic there are not many patients with complete transection of the spinal cord, there are a large number of patients with chronic lesions that do not have effective treatments. The objective of experimental treatment remains 1) to promote axonal regeneration and 2) a possible treatment for chronic stages of spinal cord injury, since most of them are directed to acute and sub-acute lesions. In the text, we expand a little more on the importance of this.

Having said all of this, all experiments were performed “according to the book”, a big study group of animals (sixty-two) was used in this work, which makes it statistically robust. At least 12 animals were allocated to each group (SCI, SCI + PPN, etc). Perhaps expanding discussion about previous works on transplantation of PPNs in SCI would have strengthened the manuscript.

We thank the author for his/her kind remarks, and hope to have addressed all concerns satisfactorily.

Round 2

Reviewer 1 Report

Comments and Suggestions for Authors

Thanks. Nothing to add.

Reviewer 2 Report

Comments and Suggestions for Authors

The authors have addressed all my concerns and questions in the revised version of the text.